# Breaking the Quadratic Barrier:
# Robust Cardinality Sketches for Adaptive Queries

**Edith Cohen** [1 2]  **Mihir Singhal** [3 1]  **Uri Stemmer** [2 1]

## Abstract

Cardinality sketches are compact data structures that efficiently estimate the number of distinct elements across multiple queries while minimizing storage, communication, and computational costs. However, recent research has shown that these sketches can fail under *adaptively chosen queries*, breaking down after approximately $\tilde{O}(k^2)$ queries, where $k$ is the sketch size. In this work, we overcome this *quadratic barrier* by designing robust estimators with fine-grained guarantees. Specifically, our constructions can handle an *exponential number of adaptive queries*, provided that each element participates in at most $\tilde{O}(k^2)$ queries. This effectively shifts the quadratic barrier from the total number of queries to the number of queries *sharing the same element*, which can be significantly smaller. Beyond cardinality sketches, our approach expands the toolkit for robust algorithm design.

## 1. Introduction

When dealing with massive datasets, compact summary structures (known as sketches) allow us to drastically reduce storage, communication, and computation while still providing useful approximate answers.

Cardinality sketches are specifically designed to estimate the number of distinct elements in a query set (Flajolet & Martin, 1985; Flajolet et al., 2007; Cohen, 1997; Alon et al., 1999; Bar-Yossef et al., 2002; Kane et al., 2010; Cohen, 2015; Blasiok, 2020). For a ground set $[n]$ of keys, a sketch is defined by a *sketching map* $S$ that maps subsets

[1]Google Research [2]School of Computer Science, Tel Aviv University, Israel [3]School of Computer Science, UC Berkeley, Berkeley, CA, USA. Correspondence to: Edith Cohen <edith@cohenwang.com>, Mihir Singhal <mihir.a.singhal@gmail.com>, Uri Stemmer <u@uri.co.il>.

*Proceedings of the $42^{nd}$ International Conference on Machine Learning*, Vancouver, Canada. PMLR 267, 2025. Copyright 2025 by the author(s).

$V \subset [n]$ to their sketches $S(V)$, and an *estimator* that processes the sketch $S(U)$ and returns an approximation of the cardinality $|U|$. An important property of sketching maps is *composability*: The sketch $S(U \cup V)$ of the union of two sets $U$ and $V$ can be computed directly from the sketches $S(U)$ and $S(V)$. Composability is crucial for most applications, particularly in distributed settings where data is generated and analyzed across multiple locations and times. To intuit why composition is necessary, note that it is not possible to obtain a cardinality estimate of $U \cup V$ from separate cardinality estimates of $U$ and $V$, since the sets may or may not overlap (the same key might occur in both sets). Therefore we must compose the sketches before applying an estimator.

It is known that any sublinear composable cardinality sketch that is statistically guaranteed for all inputs must be randomized (Kane et al., 2010). By this, we mean that the sketching map cannot be predetermined and must instead be sampled from a distribution. The guarantees are expressed in terms of the sketch size $k$: for *any* sequence of $2^{O(k)}$ queries, with *high probability over the sampling of the map*, all queries are estimated within a small relative error.

Cardinality sketches are extensively used in practice. *MinHash sketches* are composable sketches based on hash mappings of keys to priorities, where the sketch of a set is determined by the minimum priorities of its elements (Flajolet & Martin, 1985; Flajolet et al., 2007; Cohen, 1997; Broder, 2000; Rosén, 1997; Cohen, 1997; Broder, 1997; Bar-Yossef et al., 2002)[1]. Most practical implementations[2] use MinHash sketches of various types, particularly bottom-$k$ and HyperLogLog sketches (Flajolet et al., 2007; Heule et al., 2013). In many applications, a sketch is computed for each part of the data that reside in different locations and occurs in different time periods. The original data is often discarded. The estimates are used for localized analysis, where cardinality estimates are computed for disjoint datasets, but also for analysis that spans multiple locations and time horizons, which requires composability. To ensure composability, *the same map must be used to sketch all*

---

[1]For a survey see (Cohen, 2008; 2023).

[2]See, e.g., (Apache Software Foundation, Accessed: 2024; Google Cloud, Accessed: 2024).

*queries*, and the guarantees of $2^{O(k)}$ queries hold provided that the queries *do not depend on the sampled sketching map*, which is known as the *non-adaptive setting*.

## 1.1. The adaptive setting

In the adaptive setting, we assume that the sequence of queries may be chosen adaptively based on previous interactions with the sketch. This arises naturally when a feedback loop causes queries to depend on prior outputs.

The main challenge in the adaptive setting (compared to the non-adaptive setting) is that the queries become *correlated with the internal randomness of the sketch*. This would not be an issue if the sketching algorithm were deterministic, but unfortunately, randomness is necessary.

Specifically, we consider a black-box interaction model, where the sketching map $S$ is sampled once at the beginning and then fixed but hidden. The responder answers queries using this fixed map, but the map itself is not revealed. The adversary interacts with the sketch by submitting queries $U$ and observing the corresponding cardinality estimates that the responder computes from $S(U)$, and may adapt future queries based on previous outputs. This captures realistic settings where the sketch is reused across queries, and its internal randomness can potentially leak over time.

Hassidim et al. (2020) presented a *generic robustness wrapper* that transforms a non-robust randomized sketch into a more robust one. Informally, this wrapper uses *differential privacy* (Dwork et al., 2006) to obscure the internal randomness of the sketching algorithm, effectively breaking correlations between the queries and the internal randomness.

In more detail, to support $t$ adaptive queries, the wrapper of Hassidim et al. (2020) maintains approximately $\sqrt{t}$ independent copies of the non-robust sketch and answers each query by querying all (or some) of these sketches and aggregating their responses. As Hassidim et al. (2020) showed, this results in a more robust (and composable) sketch, that can support $t$ queries in total at the cost of increasing the space complexity by a factor of $\approx \sqrt{t}$. Instantiating this wrapper with classical (non-robust) cardinality sketches results in a sketch for cardinality estimation that uses space $k \approx \sqrt{t}/\alpha^2$, where $\alpha$ is the accuracy parameter.

**Lower bounds.** Lower bounds on robustness are established by designing *attacks* in the form of adaptive sequences of queries. The objective of an attack is to force the algorithm to fail. An attack is more efficient if it causes the algorithm to fail using a smaller number of adaptive queries. We refer to the number of queries in the attack as the *size* of the attack, which is typically a function of the sketch size $k$. Some attacks are *tailored* to a particular estimator, while others are *universal* in the sense that they apply to *any*

estimator.

For cardinality sketches, Reviriego & Ting (2020) and Paterson & Raynal (2021) constructed $\tilde{O}(k)$-size attacks on the popular HLL sketch with its standard estimator. Ahmadian & Cohen (2024) constructed $\tilde{O}(k)$-size attacks for popular MinHash sketching maps with their standard estimators, as well as a $\tilde{O}(k^2)$-size universal attacks. Gribelyuk et al. (2024) presented polynomial-size universal attacks on all linear sketching maps for cardinality estimation. Finally, Cohen et al. (2024) presented optimal $\tilde{O}(k^2)$-size universal attacks on broad classes of union-composable or linear sketching maps. [3]

To summarize, there are, essentially, matching upper and lower bounds of $t = \tilde{\Theta}(k^2)$ on the number of adaptive cardinality queries that can be approximated using a sketch of size $k$. The upper bound is obtained from the generic wrapper of Hassidim et al. (2020), while the lower bound arises from the universal attack of Cohen et al. (2024). We refer to this limitation as the *quadratic barrier*.

## 1.2. Per-key participation

Nevertheless, we can still hope for stronger *data-dependent* guarantees, and since cardinality sketches are widely used in practice, achieving this under realistic conditions is important. Specifically, we seek common properties of input queries such that, if these properties hold, we can guarantee accurate processing of $t \gg k^2$ adaptive queries.

We consider a parameter $r$ that is the *per-key participation* in queries. Current attack constructions are such that most keys are involved in a large number of the queries and therefore $r \approx t$. However, in many realistic scenarios, the majority of keys participate in only a small number of queries and $r \ll t$. This pattern emerges when the distribution over the key domain shifts over time. For instance, the popularity of watched videos or browsed webpages can change over time, leading to a changing set of access frequencies of keys. Additionally, even when the query distribution is fixed, this pattern is consistent with Pareto-distributed frequencies, where a small fraction of keys (the "heavy hitters") appear in most queries, while most keys appear in only a limited number of queries. We therefore pose the following question:

**Question 1.1.** Can we shift the quadratic barrier from the total number of queries $t$ to the typically much smaller parameter $r$, that is, can we design a robust (and composable) sketch of size $k \approx \sqrt{r}$ instead of $k \approx \sqrt{t}$?

---

[3]These attacks apply to any sketching map that satisfies some certain reasonability conditions and suggest that the quadratic bound is likely to apply generally.

## 1.3. Results overview

We provide an affirmative answer to Question 1.1. Specifically, we design a sketch and estimator capable of handling an exponential number of adaptive queries provided that each key participates in at most $r = \tilde{O}(k^2)$ queries. We further provide an extension which maintain the guarantee even if this condition fails for a small fraction of the keys in each query.

**Reformulating the robustness wrapper (Section 3).** As we mentioned, the generic wrapper of Hassidim et al. (2020) transforms non-robust sketches into more robust ones by obscuring their internal randomness using differential privacy. This effectively "reduces" the problem of designing a robust sketch to that of designing a suitable differentially private aggregation procedure. Our first contribution is to reformulate this wrapper so that the reduction is not to differential privacy, but rather to the problem of *adaptive data analysis (ADA)*.

In the ADA problem, we get an input dataset $S$ sampled from some unknown distribution $\mathcal{D}$, and then need to answer a sequence of *adaptively chosen statistical queries (SQ)* w.r.t. $\mathcal{D}$. This problem was introduces by Dwork et al. (2015) who showed that it is possible to answer $\approx |S|^2$ statistical queries efficiently. The application of differential privacy as a tool for the ADA problem predated its application for robust data structures.

Our reformulated wrapper has two benefits: (1) It allows us to augment the generic wrapper with the granularity needed to address Question 1.1, whereas the existing wrapper lacks this flexibility. (2) Even though our construction ultimately solves the ADA problem using differential privacy, other known solutions to the ADA problem now exist, and it is conceivable that future applications might need to leverage properties of these alternative solutions.

To this end, in Section 3 we introduce a tool: an SQ framework with fine-grained generalization guarantees. This framework addresses an analogous version of Question 1.1 for the ADA problem, where $k$ represents the sample size and $r$ is the maximum number of query predicates that are satisfied by a given key $x$. To adapt this tool for sketching, we need to represent the randomness determining the sketching map as a sample from a product distribution and express the query response algorithm in terms of appropriate statistical queries. This tool generalizes the analysis of the robust version of CountSketch introduced by Cohen et al. (2022a).

**Robust estimators for bottom-$k$ sketch.** We use our fine-grained SQ framework in order to design a robust version for the popular bottom-$k$ MinHash cardinality sketch (Rosén, 1997; Cohen, 1997; Broder, 1997; Bar-Yossef et al., 2002). The randomness in the bottom-$k$ sketch corresponds to

a map from keys to i.i.d. random priorities. The sketch BkSketch($V$) of a subset $V$ includes the $k$ keys with lowest priorities and their priority values. The standard cardinality estimator for this sketch returns a function of the highest priority in the sketch, which is a sufficient statistic for the cardinality. A sketch size of $k = \tilde{\Omega}(\alpha^{-2})$ yields with high probability a relative error of $1 \pm \alpha$. The standard estimator, however, can be compromised using an attack with $t = \tilde{O}(k)$ queries (Ahmadian & Cohen, 2024). We present two robust estimators for the bottom-$k$ sketch that are analyzed using our fine-grained SQ framework:

- **Basic Robust Estimator (Section 4).** We present a *stateless* estimator and show that for $\alpha \in (0, 1)$ and $r = \tilde{\Omega}(k^2\alpha^4)$, all estimates are accurate with high probability provided that all keys participate in no more than $r$ query sketches. In particular, since it always holds that $r \geq t$, this implies a guarantee of $t = \tilde{\Omega}(k^2\alpha^4)$ on the number of adaptive queries. Note that the sketch size "budget" of $k$ can be used to trade off accuracy and robustness.

- **Tracking Robust Estimator (Section 5).** We present another estimator that tracks the exposure of keys based on their participation in query sketches. Once a limit of $r$ is reached, the key is deactivated and is not used in future queries. The tracking estimator allows for smooth degradation and continues to be accurate as long as at most an $\alpha$ fraction of entries in the sketch are deactivated. Note that the tracking state is maintained by the query responder (server-side) and does not effect the computation or size of the sketch.

**Experiments.** Finally, in Section 6, we demonstrate the benefits of our fine-grained analysis using simulations on query sets sampled from uniform and Pareto distributions and observe $12\times$ to $100\times$ gains.

## 1.4. Additional related work

The adaptive setting has been studied extensively across multiple domains, including statistical queries (Freedman, 1983; Ioannidis, 2005; Lukacs et al., 2009; Hardt & Ullman, 2014; Dwork et al., 2015; Bassily et al., 2021), sketching and streaming algorithms (Mironov et al., 2008; Hardt & Woodruff, 2013; Ben-Eliezer et al., 2021b; Hassidim et al., 2020; Woodruff & Zhou, 2021; Attias et al., 2021; Ben-Eliezer et al., 2021a; Cohen et al., 2022b; 2023; Ahmadian & Cohen, 2024), dynamic graph algorithms (Shiloach & Even, 1981; Ahn et al., 2012; Gawrychowski et al., 2020; Gutenberg & Wulff-Nilsen, 2020; Wajc, 2020; Beimel et al., 2021), and machine learning (Szegedy et al., 2013; Goodfellow et al., 2014; Athalye et al., 2018; Papernot et al., 2017).

**Lower bounds.** For the ADA problem, Hardt & Ullman

(2014); Steinke & Ullman (2015) designed a quadratic-size universal attack, using Fingerprinting Codes (Boneh & Shaw, 1998). Hardt & Woodruff (2013) designed a polynomial-size universal attack on any linear sketching map for $\ell_2$ norm estimation. Cherapanamjeri & Nelson (2020) constructed an $\tilde{O}(k)$-size attack on the Johnson Lindenstrauss Transform with the standard estimator. Ben-Eliezer et al. (2021b) presented an $\tilde{O}(k)$-size attack on the AMS sketch (Alon et al., 1999) with the standard estimator. Cohen et al. (2022b) presented an $\tilde{O}(k)$-size attack on Count-Sketch (Charikar et al., 2002) with the standard estimator. Cohen et al. (2023) presented $\tilde{O}(k^2)$ size universal attack on the AMS sketch (Alon et al., 1999) for $\ell_2$ norm estimation and on Count-Sketch (Charikar et al., 2002) (for heavy hitter or inner product estimation).

## 2. Preliminaries

### 2.1. DP tools: linear queries with per-unit charging

Differential privacy (Dwork et al., 2006) (DP) is a Lipschitz-like stability property of algorithms, parametrized by $(\varepsilon, \delta)$. Two datasets $\boldsymbol{x}, \boldsymbol{x}' \in X^n$ are *neighboring* if they differ in at most one entry. Two probability distributions $\mathcal{D}$ and $\mathcal{D}'$ satisfy $\mathcal{D} \approx_{\varepsilon,\delta} \mathcal{D}'$ if and only if for any measurable set of events $E$, $\Pr_D(E) \le e^\varepsilon \Pr_{\mathcal{D}'}(E) + \delta$ and $\Pr_{\mathcal{D}'}(E) \le e^\varepsilon \Pr_{\mathcal{D}}(E) + \delta$. A randomized algorithm $A$ is $(\varepsilon, \delta)$-*DP* if for any two neighboring inputs $\boldsymbol{x}$ and $\boldsymbol{x}'$, $A(\boldsymbol{x}) \approx_{\varepsilon,\delta} A(\boldsymbol{x}')$. DP algorithms compose in the sense that multiple applications of a DP algorithm to the dataset are also DP (with composed parameters).

Given a dataset $\boldsymbol{x} := (x_1, \dots, x_n) \in X^n$ of items from domain $X$, a *counting query* is specified by a predicate $f : [n] \times X \to [0, 1]$ and has the form $f(\boldsymbol{x}) := \sum_{i \in [n]} f(i, x_i)$. For $\varepsilon > 0$, an algorithm that returns a noisy count $\hat{f}(\boldsymbol{x}) := f(\boldsymbol{x}) + \text{Lap}[1/\varepsilon]$, where Lap is the Laplace distribution, satisfies $(\varepsilon, 0)$-DP. When multiple such tests are performed over the same dataset, the privacy parameters compose to $(r\varepsilon, 0)$-DP or alternatively to $(\sqrt{2r \log(1/\delta)}\varepsilon + r\varepsilon^2, \delta)$-DP for any $\delta > 0$.

The Sparse Vector Technique (SVT) (Dwork et al., 2009; Roth & Roughgarden, 2010; Hardt & Rothblum, 2010; Vadhan, 2017) is a privacy analysis technique for a situation when an adaptive sequence of threshold tests on counting queries is performed on the same sensitive dataset $\boldsymbol{x}$. Each test $\text{AboveThreshold}_\varepsilon(f, T)$ is specified by a predicate $f$ and threshold value $T$. The result is the noisy value $\hat{f}(\boldsymbol{x})$ if $\hat{f} > T$ and is $\perp$ otherwise. The appeal of the technique is a privacy analysis that only depends on the number $r$ of queries for which the test result is positive.

We will use here an extension of (a stateless version of) SVT, described in Algorithm 1, that improves utility for

the same privacy parameters (Kaplan et al., 2021; Feldman & Zrnic, 2021; Cohen et al., 2022a; Cohen & Lyu, 2023b). The algorithm maintains a set $A$ of *active* indices that is initialized to all of $[n]$ and maintains charge counters $(C_i)_{i \in [n]}$, initialized to 0. For each query $(h, T)$, the response is $\text{AboveThreshold}_\varepsilon^A(h, T)$ test result that is $\hat{f} := \sum_{i \in A} h(i, x_i) + \text{Lap}[1/\varepsilon]$ if $\hat{f} > T$ and is $\perp$ otherwise. Note that $\text{AboveThreshold}^A$ evaluates the query only over active indices. For each query with a positive (above) response, the algorithm increases the charge counts on all the indices that contributed to the query, namely, $h(i, x_i) = 1$. Once $C_i = r$, index $i$ is removed from the active set $A$ of indices.

The appeal of Algorithm 1 is a fine-grained analysis that can only result in an improvement – the privacy bounds have the same dependence on the parameter $r$, that in the basic approach bounds the total number $t$ of tests with positive outcomes and in the fine-grained one bounds the (potentially much smaller) per-index participation in such tests:

**Theorem 2.1** ((Cohen & Lyu, 2023a) Privacy of Algorithm 1). *For any $\varepsilon < 1$ and $\delta \in (0, 1)$, Algorithm 1 is $(O(\sqrt{r \log(1/\delta)}\varepsilon), 2^{-\Omega(r)} + \delta)$-DP (see Theorem A.1 for more precise expressions).*

---

**Algorithm 1:** Linear Queries with Individual Privacy Charging

**Input:** Sensitive data set $(x_1, \dots, x_n) \in X^n$; privacy budget $r > 0$; Privacy parameter $\varepsilon > 0$.

**foreach** $i \in [n]$ **do** $C_i \leftarrow 0$    // Initialize counters

$A \leftarrow [n]$    // Initialize the active set

**Function** $\text{AboveThreshold}_\varepsilon^A(h, T)$

  // AboveThreshold query

  **Input:** predicate $h : [n] \times X \to \{0, 1\}$ and threshold $T \in \mathbb{R}$

  $\hat{h} \leftarrow \left(\sum_{i \in A} h(i, x_i)\right) + \text{Lap}(1/\varepsilon)$    // Laplace noise

  **if** $\hat{h} \ge T$ **then**    // Test against threshold

    **foreach** $i \in A$ *such that* $h(i, x_i) = 1$ **do**

      $C_i \leftarrow C_i + 1$

      **if** $C_i = r$ **then** $A \leftarrow A \setminus \{i\}$

    **return** $\hat{h}$

  **else**

    **return** $\perp$

**on input** $(f, T)$    // Main Loop: process queries

  **return** $\text{AboveThreshold}_\varepsilon^A(f, T)$

---

### 2.2. ADA tools

The generalization property of differential privacy applies when the dataset $\boldsymbol{x}$ is sampled from a distribution. It states that if a predicate $h$ is selected in a way that preserves the privacy of the sampled points then we can bound its generalization error: the count over $\boldsymbol{x}$ is not too far from

the expected count when we sample from the distribution. We will use the following variant of the cited works (see Appendix B for a proof):

**Theorem 2.2** (Generalization property of DP (Dwork et al., 2015; Bassily et al., 2021; Feldman & Steinke, 2017)). *Let $\mathcal{A} : X^n \to 2^X$ be an $(\varepsilon, \delta)$-differentially private algorithm that operates on a dataset of size $n$ and outputs a predicate $h : X \to \{0, 1\}$. Let $\mathcal{D} = D_1 \times \cdots D_n$ be a product distribution over $X^n$, let $\boldsymbol{x} = (x_1, \ldots, x_n) \sim \mathcal{D}$ be a sample from $\mathcal{D}$, and let $h \leftarrow \mathcal{A}(\boldsymbol{x})$. Then for any $T \geq 1$ it holds that*

$$\Pr_{\substack{\boldsymbol{x} \sim \mathcal{D}, \\ h \leftarrow \mathcal{A}(\boldsymbol{x})}} \left[ e^{-2\varepsilon} \underset{\boldsymbol{y} \sim \mathcal{D}}{\mathsf{E}} h(\boldsymbol{y}) - h(\boldsymbol{x}) > \frac{4}{\varepsilon} \log(T+1) + 2Tn\delta \right] < \frac{1}{T},$$

$$\Pr_{\substack{\boldsymbol{x} \sim \mathcal{D}, \\ h \leftarrow \mathcal{A}(\boldsymbol{x})}} \left[ h(\boldsymbol{x}) - e^{2\varepsilon} \underset{\boldsymbol{y} \sim \mathcal{D}}{\mathsf{E}} h(\boldsymbol{y}) > \frac{4}{\varepsilon} \log(T+1) + 2Tn\delta \right] < \frac{1}{T},$$

*where $h(\boldsymbol{y})$ denotes the total value of $h$ over elements of $\boldsymbol{y}$.*

When applying Theorem 2.2, we will assume that $h$ also takes the index $i$ as an argument (so, we write $h(i, x_i)$ instead of $h(x_i)$). This is equivalent because we can replace $D_i$ with a distribution that samples the tuple $(i, x_i)$ for $x_i \sim D_i$.

## 3. ADA with Fine-Grained Analysis

We now consider a variation of the ADA framework where we sample a dataset $\boldsymbol{x} \sim \mathcal{D}$ from a product distribution $\mathcal{D}$ and then process adaptive linear threshold queries as in Algorithm 1 over the dataset $\boldsymbol{x}$. The benefit of this is obtaining bounds in terms of the per-key participation in queries (that is the number of queries $h$ for which $h(i, x_i) = 1$), which is always lower than the total number of queries. Moreover, the approach tolerates a small fraction of deactivated indices in each query, which simply contribute proportionally to the error.

We bound the error due to generalization and sampling and due to the privacy noise and the deactivation of keys that reached the charging limit $r$:

**Lemma 3.1** (Generalization and sampling error bound). *Let $\mathcal{D} = D_1 \times \cdots \times D_n$ be a product distribution over $X^n$. Let $\boldsymbol{x} \sim \mathcal{D}$ be a sampled dataset. Let $\alpha, \beta > 0$ be sufficiently small (i.e., smaller than some absolute constant). Consider an execution of Algorithm 1 on dataset $\boldsymbol{x}$ with $m$ adaptive queries, parameter $r \gg \log(n/\beta)$, and*

$$\varepsilon_0 := \frac{\alpha}{4\sqrt{r \log(n/\beta)}}.$$

*Then it holds that with probability at least $1 - \beta$, for all of the $m$ query predicates $h$,*

$$\left| \underset{\boldsymbol{y} \sim \mathcal{D}}{\mathsf{E}}[h(\boldsymbol{y})] - h(\boldsymbol{x}) \right| < \alpha \cdot \underset{\boldsymbol{y} \sim \mathcal{D}}{\mathsf{E}}[h(\boldsymbol{y})] + O\left( \frac{\log(m/\beta)}{\alpha} \right).$$

*Proof.* The first claim of the sampling and generalization error, follows from Theorem 2.1 and Theorem 2.2

For the privacy parameters in Theorem 2.1 we set $\delta = \beta/n^2$ and obtain $\varepsilon = \sqrt{r \log(n^2/\beta)} \cdot \varepsilon_0 < \alpha/2\sqrt{2}$. From Theorem 2.2 we get that, for each query $h$, the additive error $|\mathsf{E}_{\boldsymbol{y} \sim \mathcal{D}} h(\boldsymbol{y}) - h(\boldsymbol{x})|$ is at most

$$(e^{2\varepsilon} - 1) \cdot \underset{\boldsymbol{y} \sim \mathcal{D}}{\mathsf{E}} h(\boldsymbol{y}) + \frac{4}{\varepsilon} \log(T+1) + 2Tn\delta,$$

with probability at least $1 - 2/T$. Note that we have $e^{2\varepsilon} - 1 < \alpha$. Thus, setting $T = 2m/\beta$ (so that a union bound over all queries gives a failure probability of $1 - \beta$), the result follows. $\square$

**Claim 3.2** (Noise and deactivation error bounds). *Under the conditions of Lemma 3.1, with probability at least $1 - \beta$, for all $m$ query predicates $h$, for $\hat{h} := \sum_{i \in A} h(i, x_i) + \mathrm{Lap}(1/\varepsilon_0)$,*

$$h(\boldsymbol{x}) - \hat{h} > -\log(2m/\beta)/\varepsilon_0,$$

$$h(\boldsymbol{x}) - \hat{h} < \log(2m/\beta)/\varepsilon_0 + \sum_{i \in [n] \setminus A} h(i, x_i).$$

*Proof.* Each Laplace noise $\mathrm{Lap}(1/\varepsilon_0)$ is bounded by $\pm \log(2m/\beta)/\varepsilon_0$ with probability at least $1 - \beta/m$, so by a union bound, with probability at least $1 - \beta$, it is bounded as such for all queries, and the result follows immediately. $\square$

The total error of $\hat{h}$ with respect to the expectation $\mathsf{E}_{\boldsymbol{y} \sim \mathcal{D}} h(\boldsymbol{y})$ is bounded by the sum of errors in Lemma 3.1 and Claim 3.2:

**Corollary 3.3.** *For some constants $c_1, c_2 > 0$, under the conditions of Lemma 3.1, with probability at least $1 - \beta$, for all $m$ queries $h$,*

$$-\Delta < \underset{\boldsymbol{y} \sim \mathcal{D}}{\mathsf{E}}[h(\boldsymbol{y})] - \hat{h} < \Delta + \sum_{i \in [n] \setminus A} h(i, x_i),$$

*where*

$$\Delta = \alpha \cdot \underset{\boldsymbol{y} \sim \mathcal{D}}{\mathsf{E}}[h(\boldsymbol{y})] + O(\alpha^{-1} \sqrt{r} \log^{3/2}(mn/\beta)).$$

We can apply this fine-grained ADA to analyze the robustness of randomized data structures (or algorithms) that sample randomness $\boldsymbol{\rho}$ and process adaptive queries $M_i$ that depend on the interaction till now and the randomness $\boldsymbol{\rho}$. To do so, we need to specify a product distribution $\mathcal{D} = D_1 \times D_2 \times \cdots \times D_n$ so that

1. The distribution of $\boldsymbol{\rho}$ is $\mathcal{D}$.

2. The queries in the original problem can be specified in terms of linear queries over $\boldsymbol{\rho}$ and have statistical guarantees of utility over $\boldsymbol{\rho} \sim \mathcal{D}$.

---

**Algorithm 2:** Bottom-$k$ Cardinality Sketch and Standard Estimator

Sample $\rho_i \sim U[0, 1]$ for $i \in [n]$. `// Randomness for the Sketching Map`

**function** `BkSketch`$_{\boldsymbol{\rho}}(V)$   `// Bottom-k sketching map using ρ`

  **Input:** Set $V \subset [n]$
  **if** $|V| \leq k$ **then**
    | **return** $\{(i, \rho_i) \mid i \in V\}$
  **else**
    | **return** $\{(i, \rho_i) \mid i \in V, \rho_i < \boldsymbol{\rho}_{(k),V}\}$ `// where`
         `ρ`$_{(k),V}$ `is the k`th `smallest in the multiset {ρ`$_i$ ` | i ∈ V}`

**function** `StdEst`$_k(S)$   `// Standard estimator`

  **Input:** A bottom-$k$ sketch $S$
  **if** $|S| < k$ **then**
    | **return** $|S|$
  **else**
    | $\tau \leftarrow \max_{(i, \rho_i) \in S} \rho_i$   `// k`th `smallest ρ`$_i$
    | **return** $(k - 1)/\tau$

**on input** $S$   `// Main Loop`
  | **return** `StdEst`$_k(S)$

---

**Algorithm 3:** Basic Robust Cardinality Estimator

**function** `RobustEst`$_k(S)$   `// Estimate |V| from BkSketch`$_{\boldsymbol{\rho}}(V)$

  **Input:** A bottom-$k$ sketch $S$, $\alpha \in (0, 0.5)$
  **if** $|S| < k$ **then**  `// Return exact value when ≤ k`
    | **return** $|S|$
  **else**
    | $\tau \leftarrow k/2n, T = (1 - \alpha)k$
    | **while** $(\tau < 1)$ *and*
      $(\tilde{h} \leftarrow \sum_{(i, \rho_i) \in S} \mathbf{1}_{(\rho_i < \tau)} + \text{Lap}(1/\varepsilon_0)) < T$
    | **do**
      | $\tau \leftarrow (1 + \alpha/4)\tau$
    | **return** $T/\tau$

**Input:** Parameters $k, r, n \geq 1$ and $\alpha, \beta > 0$
$\varepsilon_0 \leftarrow \frac{\alpha/8}{4\sqrt{r \log(n/(\beta/4))}}$ `// as Lemma 3.1 with` $\frac{\alpha}{8}$, $\frac{\beta}{4}$
**on input** $S$   `// Main Loop`
  | **return** `RobustEst`$_{k,r}(S)$

---

When applying this to sketching maps which do not contain all the information of $\boldsymbol{\rho}$, we will need to ensure that the linear queries we use can be evaluated over the sketch.

## 4. The Bottom-$k$ Cardinality Sketch

### 4.1. Sketch and standard estimator

The bottom-$k$ cardinality sketch and standard estimator are described in Algorithm 2. Let the ground set of keys be $[n]$. We sample a vector of random values $\boldsymbol{\rho} \sim [0, 1]^n$. That is, for each key $i \in [n]$ there is an associated i.i.d. $\rho_i \sim \mathcal{D}$. The vector $\boldsymbol{\rho}$ specifies the bottom-$k$ sketching map `BkSketch`$_{\boldsymbol{\rho}}(V)$ that maps a set $V \subset [n]$ to its sketch. The sketch consists of the pairs $(i, \rho_i)$ for the $k$ values of $i \in V$ such that $\rho_i$ is smallest. When $|V| \leq k$, the sketch contains all elements of $V$. Note that, though $n$ and also $|V|$ can be very large, the size of the sketch is at most $k$. This sketching map is clearly composable.[4]

For a query set $V \subset [n]$, we apply an estimator to the sketch $S := \text{BkSketch}_r(V)$ to obtain an estimate of the cardinality of $V$. The standard estimator $\text{StdEst}(S)$ computes $\tau$ which is the $k$th order statistics of the $\rho_i$ values in the sketch, which is a sufficient statistic of the cardinality. It then returns the value $(k - 1)/\tau$. The estimate is unbiased, has variance at most $|V|/(k - 2)$, and an exponential tail (see e.g. (Cohen, 2015)). This standard estimator is know

---
[4]The analysis uses a common assumption of full i.i.d. randomness in the specification of the sketching maps. Note that $O(\log k)$ bits of representation are sufficient. Implementations use pseudorandom hash maps $i \mapsto \rho_i$.

---

to optimally use the information in the sketch $S$ but can be attacked with a linear number of queries (Ahmadian & Cohen, 2024).

### 4.2. Basic robust estimator

Algorithm 4 describes a robust estimator `RobustEst` that is applied to a bottom-$k$ sketch.

We analyze this estimator under the assumption that the query set sequence $(V_j)_{j \in [t]}$ has the property that each key $i \in [n]$ appears in at most $r$ sketches in $(\text{BkSketch}_{\boldsymbol{\rho}}(V_i))_{j \in [t]}$:

$$\forall i \in [n], \sum_{j \in [t]} \mathbf{1}_{i \in \text{BkSketch}_{\boldsymbol{\rho}}(V_i)} \leq r. \tag{1}$$

Note that for this to hold it suffices that each key is included in at most $r$ query sets. That is, $\forall i \in [n], \sum_{j \in [t]} \mathbf{1}_{i \in V_j} \leq r$.

**Theorem 4.1** (Basic robust estimator guarantee). *If the query sequence in Algorithm 3 satisfies* (1) *for some* $r \gg \log(n/\beta)$, *then for a value of* $k = O(\alpha^{-2}\sqrt{r}\log^{3/2}(n/\beta))$, *every output will be* $(1 \pm \alpha)$-*accurate with probability at least* $1 - \beta$.

### 4.3. Analysis of basic robust estimator

In this section we prove Theorem 4.1.

Before we start, we make some basic assumptions on the parameters which we will use throughout the proof. First, we pick

$$k = C\alpha^{-2}\sqrt{r}\log^{3/2}(n/\beta),$$

where the constant $C$ is chosen to be sufficiently large. Furthermore, note that if $k \geq n$ then we are always storing the whole set $V$ in the sketch, so we may assume that $k < n$ (and therefore $r < n^2$ and $\alpha > 1/\sqrt{n}$).

We map Algorithm 3 to the framework of Algorithm 1, where the dataset is $\boldsymbol{\rho}$.

First, we show that we can consider the sum of $\mathbf{1}_{\rho_i < \tau}$ to be over the entire set $V$, rather than just those that are included in the bottom-$k$ sketch $S$:

**Lemma 4.2.** *Suppose that, in Algorithm 3, $\tilde{h} = \sum_{(i,\rho_i) \in S} \mathbf{1}_{(\rho_i < \tau)} + \mathrm{Lap}(1/\varepsilon_0))$ is replaced by $\hat{h} = \sum_{i \in V} \mathbf{1}_{(\rho_i < \tau)} + \mathrm{Lap}(1/\varepsilon_0))$ (where the two Laplace random variables are coupled to be the same value). Then, the sequence of outputs of Algorithm 3 changes with probability at most $\beta/4$.*

*Proof.* Since $S$ contains the $k$ values of $i$ such that $\rho_i$ is minimal, the only way to have $\hat{h} \neq \tilde{h}$ is to have the sum over $S$ be equal to $k$ and the sum over $V$ to be greater than $k$. The outputs may then only differ if $\tilde{h} < T$, but the probability that $k + \mathrm{Lap}(1/\varepsilon_0) < (1 - \alpha)k$ is at most $e^{-\varepsilon_0 \alpha k} < \beta/\mathrm{poly}(n)$, where the polynomial in $n$ can be made as large as we like (by setting the constant on $k$).

Now, note that the total number of queries to Algorithm 3 cannot exceed $nr \leq \mathrm{poly}(n)$ by (1). Furthermore, the total number of iterations of the while loop per call is at most $O(\alpha^{-1} \log n) = \mathrm{poly}(n)$.

The lemma follows by taking a union bound over all iterations of the while loop and over all queries to Algorithm 3. $\square$

With this lemma in mind, we will henceforth assume through this entire section that Algorithm 3 uses $\hat{h}$ instead of $\tilde{h}$, introducing a failure probability of at most $\beta/4$.

Now, we will show that the execution of Algorithm 3 can be performed via queries to Algorithm 1, rather than accessing $\boldsymbol{\rho}$ directly. Indeed, note that the counting query $\hat{h}$ takes the same form (except for the check being over $[n]$ instead of $A$) as its analog in Algorithm 1, where the query function is

$$h_{V,\tau}(i, \rho_i) := \mathbf{1}_{i \in V \wedge \rho_i < \tau}.$$

Observe that for any query sketch $S$, there is at most one positive test in Algorithm 3. Therefore, per assumption (1) on the input, each index appears in at most $r$ positive tests. Therefore, if we were to instead perform these tests using Algorithm 1, all indices would remain active and nothing would ever be removed from $A$. Thus, we would have that $A = [n]$, so indeed the values of $\hat{h}$ are identical in Algorithm 1 and Algorithm 3. Thus, Algorithm 3 can be simulated by queries to Algorithm 1, so we may apply the results of Section 3.

In order to apply Corollary 3.3, we need to first compute the expectation of $h_{V,\tau}$ on $\mathcal{D}$:

---

**Algorithm 4:** Tracking Robust Estimator

**function** TRobustEst$_{k,r}(S)$ // Robust Cardinality Estimate of $V$ from BkSketch$_{\boldsymbol{\rho}}(V)$

    **Input:** A bottom-$k$ sketch $S$
    **if** $|S| < k$ **then**   // Return exact value when $\leq k$
        | **return** $|S|$
    **else**
        $\tau \leftarrow k/2n, T \leftarrow k/4$
        **while**
        $(\tau < 1) \wedge (\tilde{h} \leftarrow \sum_{(i,\rho_i) \in S} \mathbf{1}_{(\rho_i < \tau) \wedge (C[i] < r)} + \mathrm{Lap}(1/\varepsilon_0)) < T$ **do**
            | $\tau \leftarrow (1 + \alpha/8)\tau$
        **foreach** $(i, x_i) \in S$ **do**   // Per-key tracking
            | **if** $x_i < \tau$ **then** $C[i] \leftarrow C[i] + 1$
        **return** $T/\tau$

**Input:** Parameters $k, r \geq 1$
// Initialization
$C \leftarrow \{\}$     // Dictionary with default value 0
$\varepsilon_0 \leftarrow \frac{\alpha/16}{4\sqrt{r \log(n/(\beta/4))}}$// as Lemma 3.1 with $\frac{\alpha}{16}$, $\frac{\beta}{4}$
**on input** $S$     // Main Loop
    | **return** TRobustEst$_{k,r}(S)$

---

**Claim 4.3.**

$$\mathsf{E}_{\boldsymbol{y}}[h_{V,\tau}(\boldsymbol{y})] = \tau|V|. \tag{2}$$

*Proof.* Observe that for $i \notin V$, $h_{V,\tau}(i, y_i) = 0$ for all $y_i$ and for $i \in V$, $\mathsf{E}[h_{V,\tau}(i, y_i)] = \tau$. $\square$

Finally, recall from the proof of Lemma 4.2 that the total number of iterations of the while loop (and thus the overall total number of calls to Algorithm 1) is at most $\mathrm{poly}(n)$, so in Corollary 3.3 we can take $m = \mathrm{poly}(n)$.

Now, by Corollary 3.3, we have with probability at least $1 - \beta/4$ that

$$\hat{h} < (1 + \alpha/8)\tau|V| + \alpha k/8, \tag{3}$$
$$\hat{h} > (1 - \alpha/8)\tau|V| - \alpha k/8, \tag{4}$$

where we have used that

$$\Delta = O(\alpha^{-1} \sqrt{r} \log^{3/2}(n/\beta)) < \alpha k/8,$$

by the choice of $k$. (Recall also that $[n] \setminus A$ is always empty, so the sum term in Corollary 3.3 vanishes.) We assume henceforth that this probability-$(1 - \beta/4)$ event does in fact occur.

**Proposition 4.4.** *Whenever $\tau < (1 - \alpha/2)T/|V|$, the while loop in Algorithm 3 continues to the next value of $\tau$.*

*Proof.* Note that $\alpha k/8 < \alpha T/4$ since $T > k/2$. Thus, by (3), when $\tau < (1 - \alpha/2)T/|V|$, we have $\hat{h} < (1 + \alpha/8)(1 - \alpha/2)T + \alpha T/4 < T$ (for sufficiently small $\alpha$), so we are done. $\square$

**Proposition 4.5.** *Whenever $\tau > (1+\alpha/2)T/|V|$, the while loop in Algorithm 3 terminates.*

*Proof.* Again, by (4), when $\tau > (1+\alpha/2)T/|V|$, we have $\hat{h} > (1-\alpha/8)(1+\alpha/2)T - \alpha T/4 > T$, so we are done. $\square$

Now, Proposition 4.4 ensures that the output of the algorithm is always at least $(1 - \alpha/2)|V|$. Moreover, since $\tau$ is incremented by factors of $1 + \alpha/4$, there will be some value of $\tau$ tested that is between $(1 + \alpha/2)T/|V|$ and $(1 + \alpha)T/|V|$ (note that since $|V| \geq k$, we have $(1 + \alpha)T/|V| < 1$). By Proposition 4.5, this value will cause the while loop to terminate, yielding an output that is at most $(1 + \alpha)|V|$. This completes the proof of Theorem 4.1.

When assumption (1) does not hold, that is, when some keys get *maxed* (have participated in more than $r$ query sketches), the guarantees are lost even when there are no maxed keys in the query set. The universal attack constructions of (Ahmadian & Cohen, 2024; Cohen et al., 2024) show this is unavoidable. The attack fixes a ground set $U$ and identifies keys with low priorities (these are the keys that tend to be maxed). The query sets that is $U$ with the identified keys deleted has cardinality close to $|U|$ but the estimates of RobustEst would be biased down. In the next section we introduce a tracking estimator that allows for smooth degradation in accuracy guarantees as keys get maxed.

## 5. Robust Estimator with Tracking

We next propose and analyze the estimator TRobustEst in Algorithm 3 that is an extension of RobustEst that includes tracking and deactivation of keys that appeared in the query sketches more than $r$ times. This estimator offers smooth degradation in estimate quality that depends only on the number of *deactivated* keys present in the sketch and this is guaranteed as long as there are no queries where most of the sketch is deactivated.

**Theorem 5.1** (Analysis of TRobustEst). *For a value of $k = O(\alpha^{-2}\sqrt{r}\log^{3/2}(n/\beta))$, suppose that an adaptive adversary provides at most $m$ inputs to Algorithm 4 such that the sketch of every input has at most $k/2$ deactivated keys. Then, with probability at least $1 - \beta$, for every input whose sketch has at most $\alpha k/4$ deactivated keys, the output is a $(1 + \alpha)$-approximation of the true cardinality.*

This theorem guarantees that, as long as no query has too many (more than $k/2$) deactivated keys, the results of Algorithm 4 will continue to be accurate even for queries that have a few (at most $\alpha k/4$) deactivated keys. This allows the algorithm to continue guaranteeing accuracy even if a few keys are subject to many queries. We discuss the numerical advantages of this further in Section 6. The proof is analogous to Section 4.3 and is provided in Appendix C.

## 6. Empirical Demonstration

We demonstrate the effectiveness of our fine-grained approach by comparing the number of queries that can be guaranteed compared with the baseline per-query analysis. We use synthetically generated query sets sampled from Uniform and Pareto distributions with $\alpha \in \{1.5, 2\}$ and $x_m = 1$, support size of $10^6$ and query set size of $5 \times 10^3$. For each sketch size $k$, we match a value of the parameter $r = 0.002k^2$ (with RobustEst and TRobustEst) and respectively $t = 0.002k^2$ with baseline analysis.

With RobustEst we stop as soon as there is a key that appeared in $r$ queries. With TRobustEst, we count the number of queries for which at most $10\%$ of sketch entries are deactivated and stop when there is a sketch with $50\%$ of entries deactivated. Figure 1 reports the number of queries with the baseline and fine-grained estimators. The respective gain factor is measured by the ratio of the number of queries that can be effectively answered with per-key analysis to the baseline. We observe gains of two orders of magnitude for uniformly sampled query sets. This hold even without tracking – using RobustEst. For Pareto query sets, we only TRobustEst, as tracking is necessary because some keys appear in many query sketches. We observe gains of $12\times$ for the very skewed $\alpha = 1.5$ and $40\times$ with $\alpha = 2$.

## Conclusion

Our work raises several compelling follow-up questions. While our fine-grained robust estimators are tailored to the bottom-$k$ cardinality sketch, we conjecture that similar robustness guarantees can be achieved for other MinHash-based sketches, including the $k$-partition (also known as PCSA or Stochastic Averaging) sketches (Flajolet & Martin, 1985; Flajolet et al., 2007). Achieving this would likely require extending the fine-grained ADA framework beyond its current restriction to linear queries.

Another open direction concerns robustness for other norms, settings where inputs are sparse and only a small subset of entries are *heavy*—appearing frequently across queries. When sketching methods assign independent randomness per key and each key appears in only one query, full robustness trivially holds. This scenario applies to many popular sketching methods. However, robustness becomes more subtle when keys may appear in multiple $r > 1$ queries or when only a small number of keys appear in many queries. For $\ell_2$ norm estimation with the popular AMS sketch (Alon et al., 1999), existing quadratic-size attacks succeed even when all inputs have disjoint support *except for a single shared key* (Cohen et al., 2023). The guarantees we can provide under the weaker condition that each key appears in at most $r > 1$ queries remains an open question.

Finally, we conjecture that analogous robustness results

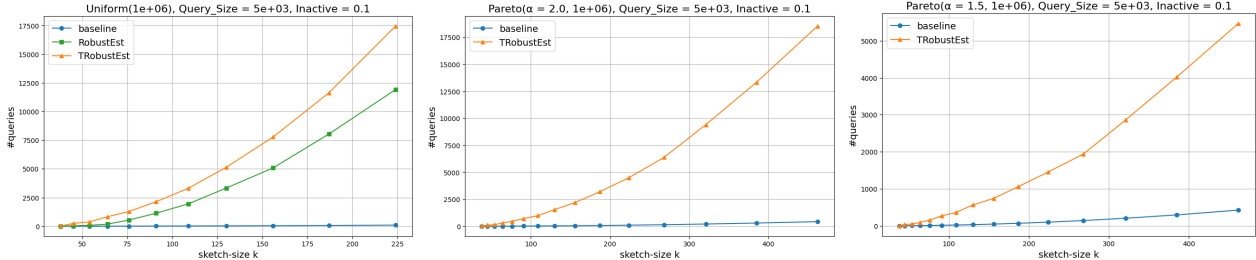

*Figure 1.* Number of guaranteed queries for sketch size $k$. The gain factor of `TRobustEst` over baseline is over two orders of magnitude with the Uniform distribution, $40\times$ for Pareto with $\alpha = 2$, and $12\times$ for Pareto with $\alpha = 1.5$.

may be achievable for a broader class of sublinear statistics—such as capping statistics (Cohen, 2018; Cohen & Geri, 2019), whose sketching techniques extend generalized cardinality sketches.

## Acknowledgments

Edith Cohen was partially supported by Israel Science Foundation (grant 1156/23). Uri Stemmer was Partially supported by the Israel Science Foundation (grant 1419/24) and the Blavatnik Family foundation.

## Impact Statement

This paper presents work whose goal is to advance the field of Machine Learning. There are many potential societal consequences of our work, none which we feel must be specifically highlighted here.

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

## A. Fine-grained SVT privacy bounds

More precise bounds for Theorem 1 and for standard SVT throught the target charging technique (TCT).

**Theorem A.1** (Privacy of Target-Charging (Cohen & Lyu, 2023a) )**.** *Algorithm 1 (and per-query SVT) satisfy the following approximate DP privacy bounds:*

$$\left( (1+\alpha)\frac{r}{q}\varepsilon, \delta^*(r,\alpha) \right), \qquad\qquad\qquad\qquad \textit{for any } \alpha > 0;$$

$$\left( \frac{1}{2}(1+\alpha)\frac{r}{q}\varepsilon^2 + \varepsilon\sqrt{(1+\alpha)\frac{r}{q}\log(1/\delta)}, \delta + \delta^*(r,\alpha) \right), \qquad \textit{for any } \delta > 0, \alpha > 0.$$

*where* $\delta^*(r,\alpha) \leq e^{-\frac{\alpha^2}{2(1+\alpha)}r}$ *and* $q = \frac{1}{e^\varepsilon + 1}$.

## B. Proof of extended generalization

Here we will prove Theorem 2.2:

**Theorem 2.2** (Generalization property of DP (Dwork et al., 2015; Bassily et al., 2021; Feldman & Steinke, 2017))**.** *Let* $\mathcal{A} : X^n \to 2^X$ *be an* $(\varepsilon, \delta)$*-differentially private algorithm that operates on a dataset of size* $n$ *and outputs a predicate* $h : X \to \{0, 1\}$*. Let* $\mathcal{D} = D_1 \times \cdots D_n$ *be a product distribution over* $X^n$*, let* $\boldsymbol{x} = (x_1, \ldots, x_n) \sim \mathcal{D}$ *be a sample from* $\mathcal{D}$*, and let* $h \leftarrow \mathcal{A}(\boldsymbol{x})$*. Then for any* $T \geq 1$ *it holds that*

$$\Pr_{\substack{\boldsymbol{x}\sim\mathcal{D}, \\ h\leftarrow\mathcal{A}(\boldsymbol{x})}} \left[ e^{-2\varepsilon} \mathop{\mathsf{E}}_{\boldsymbol{y}\sim\mathcal{D}} h(\boldsymbol{y}) - h(\boldsymbol{x}) > \frac{4}{\varepsilon}\log(T+1) + 2Tn\delta \right] < \frac{1}{T},$$

$$\Pr_{\substack{\boldsymbol{x}\sim\mathcal{D}, \\ h\leftarrow\mathcal{A}(\boldsymbol{x})}} \left[ h(\boldsymbol{x}) - e^{2\varepsilon} \mathop{\mathsf{E}}_{\boldsymbol{y}\sim\mathcal{D}} h(\boldsymbol{y}) > \frac{4}{\varepsilon}\log(T+1) + 2Tn\delta \right] < \frac{1}{T},$$

The proof is obtained by transforming the following *expectation bound* into a *high probability bound*.

**Lemma B.1** (Expectation bound (Kontorovich et al., 2022))**.** *Let* $\mathcal{B}$ *be an* $(\varepsilon, \delta)$*-differentially private algorithm that operates on* $T$ *sub-databases and outputs a predicate* $h : X \to [0, 1]$ *and an index* $t \in \{1, 2, \ldots, T\}$*. Let* $\mathcal{D} = D_1 \times \cdots D_n$ *be a product distribution over* $X^n$ *be a distribution over* $X$*, let* $\vec{\boldsymbol{x}} = (\boldsymbol{x}_1, \ldots, \boldsymbol{x}_T)$ *where every* $\boldsymbol{x}_j \sim \mathcal{D}$ *is sampled independently, and let* $(h, t) \leftarrow \mathcal{B}(\vec{\boldsymbol{x}})$*. Then,*

$$\mathop{\mathsf{E}}_{\substack{\vec{\boldsymbol{x}}\sim\mathcal{D} \\ (h,t)\leftarrow\mathcal{B}(\vec{\boldsymbol{x}})}} \left[ e^{-\varepsilon} \cdot h(\mathcal{D}) \right] - Tn\delta \quad \leq \quad \mathop{\mathsf{E}}_{\substack{\vec{\boldsymbol{x}}\sim\mathcal{D} \\ (h,t)\leftarrow\mathcal{B}(\vec{\boldsymbol{x}})}} \left[ h(\boldsymbol{x}_t) \right] \quad \leq \quad \mathop{\mathsf{E}}_{\substack{\vec{\boldsymbol{x}}\sim\mathcal{D} \\ (h,t)\leftarrow\mathcal{B}(\vec{\boldsymbol{x}})}} \left[ e^\varepsilon \cdot h(\mathcal{D}) \right] + Tn\delta.$$

(Here, we use $h(\mathcal{D})$ as shorthand to denote $\mathsf{E}_{\boldsymbol{y}\sim\mathcal{D}} h(\boldsymbol{y})$.)

*Proof.* The proof is identical to that of Lemma 3.1 of (Kontorovich et al., 2022) with $\psi = 0$, and omitting the final inequality in the last chain of inequalities. $\square$

*Proof of Theorem 2.2.* We prove the first inequality; the second follows from similar arguments. Fix a product distribution $\mathcal{D}$ on $X$. Assume towards contradiction that with probability at least $1/T$ algorithm $\mathcal{A}$ outputs a predicate $h$ such that $e^{-2\varepsilon} \cdot h(\mathcal{D}) - h(\boldsymbol{x}) > \frac{4}{\varepsilon}\log(T+1) + 2Tn\delta$. We now use $\mathcal{A}$ and $\mathcal{D}$ to construct the following algorithm $\mathcal{B}$ that contradicts Lemma B.1. We remark that algorithm $\mathcal{B}$ "knows" the distribution $\mathcal{D}$. This will still lead to a contradiction because the expectation bound of Lemma B.1 holds for *every* differentially private algorithm and *every* underlying distribution.

Observe that $\mathcal{B}$ only accesses its input through $\mathcal{A}$ (which is $(\varepsilon, \delta)$-differentially private) and the exponential mechanism (which is $(\varepsilon, 0)$-differentially private). Thus, by composition and post-processing, $\mathcal{B}$ is $(2\varepsilon, \delta)$-differentially private. Now consider applying $\mathcal{B}$ on databases $\vec{\boldsymbol{x}} = (\boldsymbol{x}_1, \ldots, \boldsymbol{x}_T)$ containing i.i.d. samples from $\mathcal{D}$. By our assumption on $\mathcal{A}$, for every $t$ we have that $e^{-2\varepsilon} \cdot h_t(\mathcal{D}) - h_t(\boldsymbol{x}_t) \geq \frac{4}{\varepsilon}\log(T+1) + 2Tn\delta$ with probability at least $1/T$. We therefore get

$$\Pr_{\substack{\vec{\boldsymbol{x}}\sim\mathcal{D} \\ \mathcal{B}(\vec{\boldsymbol{x}})}} \left[ \max_{t\in[T]} \left\{ e^{-2\varepsilon} \cdot h_t(\mathcal{D}) - h_t(\boldsymbol{x}_t) \right\} \geq \frac{4}{\varepsilon}\log(T+1) + 2Tn\delta \right] \geq 1 - (1 - 1/T)^T \geq \frac{1}{2}.$$

---

**Algorithm 5:** $\mathcal{B}$

---

**Input:** $T$ databases of size $n$ each: $\vec{x} = (x_1, \ldots, x_T)$

Define $h^0 \equiv 0$ and set $F \leftarrow \{(h^0, 1)\}$.

**for** $t = 1, \ldots, T$ **do**

$\quad$ Let $h_t \leftarrow \mathcal{A}(x_t)$, and set $F = F \cup \{(h_t, t)\}$

Sample $(h^*, t^*)$ from $F$ with probability proportional to $\exp\left(\frac{\varepsilon}{2}\left(e^{-2\varepsilon} \cdot h^*(\mathcal{D}) - h^*(x_{t^*})\right)\right)$.

**return** $(h^*, t^*)$.

---

The probability is taken over the random choice of the examples in $\vec{x}$ according to $\mathcal{D}$ and the generation of the predicates $h_t$ according to $\mathcal{B}(\vec{x})$. Thus, by Markov's inequality,

$$\mathop{\mathbb{E}}_{\substack{\vec{x} \sim \mathcal{D} \\ \mathcal{B}(\vec{x})}}\left[\max\{0, \max_{t \in [T]}\{e^{-2\varepsilon} \cdot h_t(\mathcal{D}) - h_t(x_t)\}\}\right] \geq \frac{2}{\varepsilon}\log(T+1) + Tn\delta.$$

Recall that the set $F$ (constructed in step 2 of algorithm $\mathcal{B}$) contains the predicate $h^0 \equiv 0$, and hence,

$$\mathop{\mathbb{E}}_{\substack{\vec{x} \sim \mathcal{D} \\ \mathcal{B}(\vec{x})}}\left[\max_{(h,t) \in F}\{e^{-2\varepsilon} \cdot h_t(\mathcal{D}) - h_t(x_t)\}\right] = \mathop{\mathbb{E}}_{\substack{\vec{x} \sim \mathcal{D} \\ \mathcal{B}(\vec{x})}}\left[\max\{0, \max_{t \in [T]}\{e^{-2\varepsilon} \cdot h_t(\mathcal{D}) - h_t(x_t)\}\}\right] \geq \frac{2}{\varepsilon}\log(T+1) + Tn\delta. \quad (5)$$

So, in expectation, the set $F$ contains a pair $(h, t)$ with large difference $e^{-2\varepsilon} \cdot h(\mathcal{D}) - h(x_t)$. In order to contradict the expectation bound of Lemma B.1, we need to show that this is also the case for the pair $(h^*, t^*)$ that is sampled in Step 3. Indeed, by the properties of the exponential mechanism, we have that

$$\mathop{\mathbb{E}}_{(h^*, t^*) \in_R F}\left[e^{-2\varepsilon} \cdot h^*(\mathcal{D}) - h^*(x_{t^*})\right] \geq \max_{(h,t) \in F}\{e^{-2\varepsilon} \cdot h(\mathcal{D}) - h(x_t)\} - \frac{2}{\varepsilon}\log(T+1). \quad (6)$$

Taking the expectation also over $\vec{x} \sim \mathcal{D}$ and $\mathcal{B}(\vec{x})$ we get that

$$
\begin{aligned}
\mathop{\mathbb{E}}_{\substack{\vec{x} \sim \mathcal{D} \\ \mathcal{B}(\vec{x})}}\left[e^{-2\varepsilon} \cdot h^*(\mathcal{D}) - h^*(x_{t^*})\right] &\geq \mathop{\mathbb{E}}_{\substack{\vec{x} \sim \mathcal{D} \\ \mathcal{B}(\vec{x})}}\left[\max_{(h,t) \in F}\{e^{-2\varepsilon} \cdot h(\mathcal{D}) - h(x_t)\}\right] - \frac{2}{\varepsilon}\log(T+1) \\
&\geq \frac{2}{\varepsilon}\log(T+1) + Tn\delta - \frac{2}{\varepsilon}\log(T+1) = Tn\delta.
\end{aligned}
$$

This contradicts Lemma B.1. $\qquad\square$

## C. Analysis of the tracking estimator

Here we will prove Theorem 5.1:

**Theorem 5.1** (Analysis of `TRobustEst`). *For a value of $k = O(\alpha^{-2}\sqrt{r}\log^{3/2}(n/\beta))$, suppose that an adaptive adversary provides at most $m$ inputs to Algorithm 4 such that the sketch of every input has at most $k/2$ deactivated keys. Then, with probability at least $1 - \beta$, for every input whose sketch has at most $\alpha k/4$ deactivated keys, the output is a $(1 + \alpha)$-approximation of the true cardinality.*

The analysis is analogous to Section 4.3; indeed, we will reuse most of the results from that section. We may make the same assumptions on $k, r, \alpha$ as at the start of Section 4.3. Again, we begin with an analog of Lemma 4.2:

**Lemma C.1.** *If $\tilde{h} = \sum_{(i,\rho_i) \in S} \mathbf{1}_{(\rho_i < \tau) \wedge (C[i] < r)} + \mathrm{Lap}(1/\varepsilon_0)$ is replaced by $\hat{h} = \sum_{i \in V} \mathbf{1}_{(\rho_i < \tau) \wedge (C[i] < r)} + \mathrm{Lap}(1/\varepsilon_0)$, then the outputs of Algorithm 4 change with probability at most $\beta/4$.*

*Proof.* In order for $\tilde{h}$ not to equal $\hat{h}$, the maximum value of $\rho_i$ in $S$ must be below $\tau$. However, in that case, by assumption, at most $k/2$ of these elements may be inactive, so the sum in $\tilde{h}$ is at least $k/2$. In order for the output to change in any given step, we must then have $\tilde{h} < T < \hat{h}$. However, this would require $\mathrm{Lap}(1/\varepsilon_0) < -k/4$, which has probability at most $e^{-\varepsilon_0 k/4} < m/\beta$. By a union bound (as in the proof of Lemma 4.2), we are done. $\qquad\square$

Again, we assume henceforth that Algorithm 4 uses $\hat{h}$ instead of $\tilde{h}$. We now show once again that Algorithm 4 can be simulated by calls to Algorithm 1, with the same function $h_{V,\tau}$ as in Section 4.3. Indeed, the only difference from Algorithm 1 is that $C$ can only increment elements of the sketch $S$ rather than the whole set $V$, so we need to ensure that there are never keys $i \in V$ such that $\rho_i < \tau$ and $i \notin S$. We show this, along with the analogs of Proposition 4.4 and Proposition 4.5, by induction:

**Claim C.2.** The following holds with probability at least $1 - \beta/2$. Let $d$ be the number of deactivated elements in the sketch $S$. Then, whenever $\tau < (1 - \alpha/4)T/|V|$, the while loop in Algorithm 4 continues to the next value of $\tau$, and whenever $\tau > ((1 + \alpha/4)T + d)/|V|$, it terminates. Moreover, the values in $C$ always match the values that Algorithm 1 would have.

*Proof.* We proceed by induction; suppose the statement has held true on all previous inputs and iterations of the while loop. We show that it holds on the current iteration — note that we must have $\tau < (1 + 3\alpha/8)(T + d)/|V|$ by the inductive hypothesis, since otherwise the loop would have terminated in the previous step. Recall by assumption that $d < k/2$, and we set $T = k/4$, so this means that $\tau < \frac{7}{8} \cdot k/|V|$.

We first show that on the current input, the keys $i \in V$ with $\rho_i < \tau$ are all in the sketch $S$. Indeed, by the inductive hypothesis, we may apply Lemma 3.1 on $h_{V,\tau}$:

$$|\tau|V| - h_{V,\tau}(\boldsymbol{\rho})| < \frac{\alpha}{32} \cdot \tau|V| + O(\log(m/\beta)/\alpha)$$
$$< \frac{\alpha}{32} \cdot \tau|V| + \frac{\alpha k}{8},$$

Since $\tau|V| < 7k/8$, this means that we have $h_{V,\tau}(\boldsymbol{\rho}) < k$. However, $h_{V,\tau}(\boldsymbol{\rho})$ is just the count of $i \in V$ such that $\rho_i < \tau$, so if this count is less than $k$, then all such $i \in V$ are included in the sketch (since it is a bottom-$k$ sketch).

Therefore, we have shown the second part of Claim C.2, since every value that would need to be incremented is actually in the sketch $C$. It remains to show the first part.

We now apply Corollary 3.3 (again using the inductive hypothesis that our algorithm has matched Algorithm 1), to obtain that

$$\hat{h} < (1 + \alpha/16)\tau|V| + \alpha k/16 + d, \tag{7}$$
$$\hat{h} > (1 - \alpha/16)\tau|V| - \alpha k/16, \tag{8}$$

where again, the value of $\Delta$ is at most $\alpha k/16$ by the choice of $k$, and the sum in Corollary 3.3 is bounded by the number of deactivated elements satisfying $h_{V,\tau}(i, \rho_i) = 1$, which is at most $d$ (since we just showed that all such elements are in $S$). The remainder of this proof is now identical to that of Proposition 4.4 and Proposition 4.5. $\square$

From Claim C.2, we deduce (identically to the previous analysis) that whenever $d < \alpha k/4$, the output of the algorithm is a $(1 + \alpha)$-approximation of $|V|$.

