# OpenReview forum: "Breaking the Quadratic Barrier: Robust Cardinality Sketches for Adaptive Queries"
_ICML.cc/2025/Conference — ICML 2025 poster_

### Official Review · Reviewer_qFNd · 2025-03-13

**Overall Recommendation:** 4

**Summary:**

This work revisits the problem of robust cardinality sketches for adaptive queries and provides improved results. In the classic cardinality sketch, the queries are independent of the sampled sketch, and the sketches can answer an exponential number of queries (in the sketch size $k$). In the adaptive queries setting, a query can be chosen based on the output of previous queries. Existing work shows that these sketches can fail after $\tilde{O}(k^2)$ queries.

The paper breaks this quadratic barrier. It presents an adaptive data analysis framework that builds on the generalization property of differential privacy. It shows that by limiting each element's participation to $\tilde{O}(k^2)$ times, the framework can answer an exponential number of queries. The paper further fits the Bottom-k sketch into this framework, resulting in a cardinality sketch that can answer $\Omega(k^2)$ adaptive queries.

**Claims And Evidence:**

Yes.

**Essential References Not Discussed:**

No.

**Experimental Designs Or Analyses:**

Experiments validate the theoretical findings.

**Methods And Evaluation Criteria:**

Yes.

**Other Comments Or Suggestions:**

N/A

**Other Strengths And Weaknesses:**

The paper is well presented. It introduces the problem and relevant background in a succinct and accurate manner and presents the algorithms in an organized way. The contribution is substantial and offers a clear improvement over previous work in this area.

**Questions For Authors:**

I don't have any specific questions.

**Relation To Broader Scientific Literature:**

The paper broadens the toolkit for designing robust algorithms.

**Theoretical Claims:**

Yes.

---

> ### Author Rebuttal · Authors · 2025-03-29
>
> Thank you very much for your time and comments.

---

### Official Review · Reviewer_hTkY · 2025-03-14

**Overall Recommendation:** 3

**Summary:**

Cardinality sketches support computing a small sketch/summary of a set of keys S from a universe U. The sketch supports estimating |S|. This problem is trivial without further requirements as one can merely store |S| in log(|U|) bits. A cardinality sketch thus further requires that given sketches of two sets A and B, one can compute the sketch of their union and intersection from the sketches themselves. When analysing the performance of a sketch, the authors considers how many adaptively chosen queries (cardinality estimates) that a sketch of size k can process without making a mistake. The paper designs sketches that can handle more than O(k^2) queries with sketch size k. Compared to previous work, the guarantee they give depend on the number of queries that share an element in U, instead of just the number of queries.

Specifically, they claim to design a sketch and an estimator that can handle an exponential number of adaptive queries if each key participates in at most \tilde{O](k^2) queries. They also claim to extend this result to the case where this condition fails on a small fraction of keys.

They also state that one of their contributions is arguing that one can reduce constructing robust sketches to "adaptive data analysis (ADA)" instead of reducing to differential privacy.

**Claims And Evidence:**

The claims are supported by theoretical proofs. The ones that I read seem correct. Additionally, there is a small simulation study. It is not clear to me how the simulations support their claims.

Furthermore, they claim that it is among their contributions to reduce the problem to adaptive data analysis instead of reducing it to the usual differential privacy. However, they then solve the adaptive data analysis problem using differential privacy, so it's not clear to me that their approach is as novel as claimed.

**Essential References Not Discussed:**

Not that I am aware of.

**Experimental Designs Or Analyses:**

As mentioned above, it is unclear to me what the "emperical evaluation" actually says about their results, so I cannot determine if they are valid or not.

**Methods And Evaluation Criteria:**

The comparisons to theoretical results in the litterature are appropriate.

The emperical evaluation using simulated synthetic data however, is not sufficiently explaned. Hence, it is difficult to determine if it's appropriate or not.

**Other Comments Or Suggestions:**

None

**Other Strengths And Weaknesses:**

The paper is relatively easy to follow and so are the proofs. I think the results are nice and seem like they could be useful.

The biggest weakness is the section with the simulated data and the plots, which I think would benefit from a better explanation.

**Questions For Authors:**

None

**Relation To Broader Scientific Literature:**

Their algorithm and analysis are based on the privacy of their Algorithm 1 (Cohen & Lyu 2023), and they claim to break the "quadratic barrier" given by Cohen et al. (2024).

**Theoretical Claims:**

I checked all the proofs on pages 1-8 except for the proofs of the following which I merely skimmed:
lemma 3.1, claim 3.2, corollary 3.3.

All the proofs seem correct.

---

> ### Author Rebuttal · Authors · 2025-03-29
>
> Thank you very much for your time and comments.
>
> **> The biggest weakness is the section with the simulated data and the plots, which I think would benefit from a better explanation.**
>
> We will make sure to add a more clear explanation of the empirical evaluation for the full version of the paper. To explain briefly here, the plots show the gain of our "tracking" robust estimator (Algorithm 4, which keeps track of the number of times each key is used) over our "basic" robust estimator (Algorithm 3), and over the baseline from prior work. The plots show how long the theoretical guarantee of each algorithm lasts, in the case where the query sets are drawn from some selected natural distributions. The plots demonstrate that, indeed, for these distributions, Algorithm 4 is likely to maintain its guarantee for much longer than Algorithm 3, which in turn is likely to outperform the baseline from prior work.
>
> **> Reduce the problem to ADA instead of the usual DP... then solve using DP. So it's not clear to me that their approach is as novel as claimed.**
>
> By now we know of several approaches to solve the ADA problem. DP is definitely one of the prominent approaches, but it is not the only approach. In particular, as Blanc (2023) showed, the ADA problem can be solved using "subsampling" (without adding noise to the responses directly). Even though in our paper we do use DP at the end of the day to solve the resulting ADA problem, our new reduction shows that if tomorrow someone will come up with yet another solution to the ADA problem (maybe with some additional properties / providing more fine-tuned guarantees) then this new solution could potentially be applied, via our reduction, to design new robust sketches.

---

### Official Review · Reviewer_f3xR · 2025-03-17

**Overall Recommendation:** 2

**Summary:**

The paper addresses the problem of sketching under the additional constraint that each key appears in at most $r$ queries. This setting generalizes prior work by introducing a finer-grained robustness notion based on per-key participation rather than the total number of queries. The main result establishes that, under this assumption, the required sketch size should be proportional to $\sqrt{r}$, rather than the sqrt of the number of queries that can be answered accurately. The authors provide theoretical justification for this claim and propose a robust estimation procedure based on per-key tracking.

## update after rebuttal

I believe the paper still requires substantial revision, including the overall structure of the paper, clarity of exposition, problem formulation, and experimental depth. I will maintain my original score.

**Claims And Evidence:**

The primary claim made by the paper is that the sketch size should be order of $\sqrt{r}$ rather than a sqrt of the total number of queries. This result is formalized in Theorem 4.1. The theorem appears well-supported, and the authors provide proofs and an empirical evaluation. However, the paper lacks a rigorous, self-contained problem formulation, making it difficult for those unfamiliar with sketching literature to follow the argument effectively. The definition of "query" is ambiguous in the introduction and should be clarified.

**Essential References Not Discussed:**

I am not familiar with sketching literatures.

**Experimental Designs Or Analyses:**

The experiments are conducted on toy datasets, which provide some intuition for the effectiveness of the approach but lack depth.

**Methods And Evaluation Criteria:**

The proposed methods involve a modification of bottom-$k$ sketches by introducing per-key participation tracking and a robust estimation procedure. While the techniques seem reasonable, the exposition is convoluted, making it difficult to assess the practical implications. The evaluation is based on synthetic data and demonstrates the improvement over prior methods in terms of the number of adaptive queries the sketch can support. However, a more extensive comparison with standard adaptive sketching techniques would strengthen the argument.

**Other Comments Or Suggestions:**

see the Strengths and Weaknesses section.

**Other Strengths And Weaknesses:**

A major weakness of the paper is its organization and clarity. The structure is unfriendly to readers unfamiliar with sketching. A more rigorous and formal problem definition should be included early on. The term "query" is not well-defined in its initial usage, making the motivation unclear.

**Questions For Authors:**

see the Strengths and Weaknesses section.

**Relation To Broader Scientific Literature:**

he paper situates itself within the literature on cardinality estimation and sketching but does not provide sufficient background for readers unfamiliar with the area.

**Theoretical Claims:**

Theorem 4.1 states that robustness in the per-key participation model requires a sketch size of $\sqrt{r}$. While the proof structure appears correct at a high level, I have not verified the details rigorously. Given the importance of this result, providing a clear high-level overview would be beneficial while having proof details in the appendicies.

---

> ### Author Rebuttal · Authors · 2025-03-29
>
> Thank you very much for your time and comments.
>
> **> The paper lacks a rigorous, self-contained problem formulation, making it difficult for those unfamiliar with sketching literature to follow the argument effectively. The definition of "query" is ambiguous in the introduction and should be clarified.**
>
> We will make sure to define the model more clearly in the introduction. To clarify here, the model is that there is an adversary selecting a set V (the "query"), and the algorithm receives the sketch S(V) and must compute an estimate of the cardinality |V| and return this to the adversary. This process is repeated over multiple rounds.
>
> **> a more extensive comparison with standard adaptive sketching techniques would strengthen the argument.**
>
> We will survey additional related results. Prior to our work, all existing approaches to adaptive cardinality sketching obtained worst-case bounds, supporting at most a quadratic number of queries. The blue line in our plots ("baseline") captures the best prior result, showing our improvements.
>
> **> The experiments are conducted on toy datasets, which provide some intuition for the effectiveness of the approach but lack depth.**
>
> We agree that we used synthetic datasets for our evaluations. The overall goal of this paper is to present a theoretical framework and provide proven guarantees, which we consider to be the main contribution of our work.
>
> **> The exposition is convoluted... does not provide sufficient background... Providing a clear high-level overview would be beneficial...**
>
> Thank you for the feedback. For the revised version we will make sure to make the explanations, especially in the introduction, more clear and accessible, and properly define what a query is.

---

### Decision · Program_Chairs · 2025-05-01

**Decision:**

Accept (poster)

**Comment:**

This paper presents a theoretically strong contribution that breaks the quadratic barrier in adaptive cardinality sketching by introducing a per-key participation model. The approach expands the robustness framework for sketches and offers new insights by connecting adaptive sketching to adaptive data analysis (ADA).

The submission is borderline. While Reviewer qFNd strongly supports acceptance and Reviewer hTkY finds the results promising, Reviewer f3xR raises several concerns, particularly regarding clarity, accessibility, problem formulation, and experimental depth. The weaknesses they point out include that the exposition is dense, the definition of certain key terms are delayed, and the empirical evaluation is underdeveloped.

Given the soundness of the theoretical contributions and the potential impact on robust sketching, I recommend a weak accept. Note that this decision may be overridden by the SAC/program committee depending on the available room in the program. The authors are encouraged to significantly improve the clarity of the problem setup, refine the empirical section, and address accessibility issues in the final version.